# Nobody ever questions—Polypharmacy in care homes: A mixed methods evaluation of a multidisciplinary medicines optimisation initiative

Sue Jordan[1]*, Hayley Prout[2], Neil Carter[1], John Dicomidis[3], Jamie Hayes[4], Jeffrey Round[5], Andrew Carson-Stevens[6]

1 Faculty of Health and Life Science, Swansea University, Swansea, United Kingdom, 2 Centre for Trials Research, College of Biomedical and Life Sciences, Cardiff University, Cardiff, Wales, United Kingdom, 3 Care Home Governance and National Lead Pharmacy Informatics, Pontypool, Wales, United Kingdom, 4 School of Pharmacy and Pharmaceutical Sciences, School of Medicine, Cardiff University, Cardiff, Wales, United Kingdom, 5 Institute of Health Economics, Edmonton, Alberta, Canada, 6 Division of Population Medicine, School of Medicine, Cardiff University, Cardiff, Wales, United Kingdom

* s.e.jordan@swansea.ac.uk

**Data Availability Statement:** Data are provided within the supplementary tables. Ethical restrictions have been imposed on data sharing by the NHS

## Abstract

### Background

Nurse-led monitoring of patients for signs and symptoms associated with documented 'undesirable effects' of medicines has potential to prevent avoidable harm, and optimise prescribing.

### Intervention

The Adverse Drug Reaction Profile for polypharmacy (ADRe-p) identifies and documents putative adverse effects of medicines commonly prescribed in primary care. Nurses address some problems, before passing ADRe-p to pharmacists and prescribers for review, in conjunction with prescriptions.

### Objectives

We investigated changes in: the number and nature of residents' problems as recorded on ADRe-p; prescription regimens; medicines optimisation: and healthcare costs. We explored aetiologies of problems identified and stakeholders' perspectives.

### Setting and participants

In three UK care homes, 19 residents completed the study, December 2018 to May 2019. Two service users, three pharmacists, six nurses gave interviews.

### Methods

This mixed-method process evaluation integrated data from residents' ADRe-ps and medicines charts, at the study's start and 5–10 weeks later.

Research Ethics Committee that approved this study. The data contain potentially identifying and sensitive information. The data used in this study are available to the research data community at https://zenodo.org/record/4090384#.X4hKptZFzLZ Swansea University, Swansea, UK. All proposals to view the data are subject to review by Swansea University's Research Governance department and the PI. Before any data can be accessed, approval must be given. The application process is via the Academic Lead for Research Integrity Research Engagement & Innovation Services, Swansea University and the PI or Neil Carter. Contacts: Swansea University, Swansea SA2 8PP • Tel: +44 /0 1792 606060 and 518541 or 295610 • Email: researchgovernance@swansea.ac.uk, s.e. jordan@swansea.ac.uk or n.carter@swansea.ac.uk The research instrument used in the study is available for clinical use without charge via the project website: http://www.swansea.ac.uk/adre/.

**Funding:** The study was funded by £75,000 from the Health Foundation https://www.health.org.uk/ funding-and-partnerships/funding-for-improvement-projects. Innovating for Improvement Round 6. Project title: Nurse-led intervention to minimise adverse drug reactions for older adults in care homes (ACS, SJ, JH, JR, HP). The Unique Award Reference Number is 457154, allocated August 2017. The funders played no role in study design, collection, management, analysis and interpretation of data or writing of the report. They did not play any role in the decision to submit the report for publication. They have no ultimate authority over any of these activities. There was no involvement from commercial companies.

**Competing interests:** The authors have declared that no competing interests exist.

## Results

We recruited three of 27 homes approached and 26 of 45 eligible residents; 19 completed ADRe-p at least twice. Clinical gains were identified for 17/19 residents (mean number of symptoms 3 SD 1.67, range 0–7). Examples included management of: pain (six residents), seizures (three), dyspnoea (one), diarrhoea (laxatives reduced, two), falls (two of five able to stand). One or more medicine was de-prescribed or dose reduced for 12/19 residents. ADRe administration and review cost ~£30 in staff time. ADRe-p helped carers and nurses bring residents' problems to the attention of prescribers.

## Implications

ADRe-p relieved unnecessary suffering. It supported carers and nurses by providing a tool to engage with pharmacists and prescribers, and was the only observable strategy for multi-disciplinary team working around medicines optimisation. ADRe-p improved care by: a) regular systematic checks and problem documentation; b) information transfer from care home staff to prescribers and pharmacists; c) recording changes.

## Registration

NLM Identifier NCT03955133; ClinicalTrials.gov.

## Introduction

The scale and complexity of inadvertent iatrogenic harm from both use and misuse of medicines underlie the World Health Organisation's (WHO) Third Global Patient Safety Challenge—to reduce avoidable medication-related harm by 50% by 2022 [1]. Structured, multi-faceted, multi-professional interventions have an important role in mitigating risk, notably in primary and community care, where the WHO recognises a dearth of interventions to improve patient safety. Our intervention, the Adverse Drug Reaction (ADRe) Profile [2–4], achieves this by uniting the tacit, experiential knowledge of nurses and carers with records of the patients' clinical problems and possible causes, in a form that can be shared within the multidisciplinary team [2–7] (Fig 1).

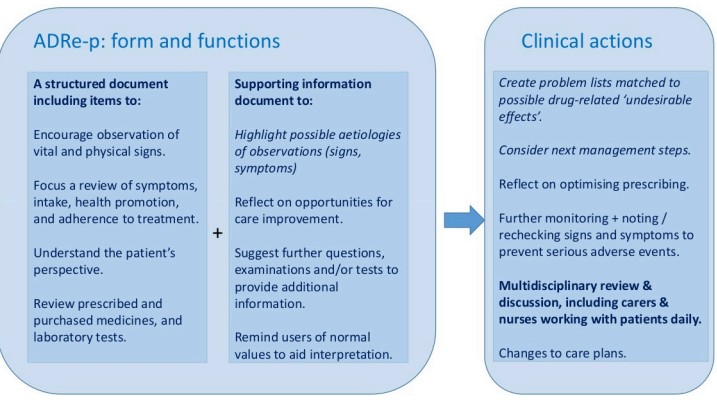

**Fig 1. How ADRe works.**

## Background

Identification of medicine-related incidents as the most prevalent source of unsafe primary care in Wales and England underlines the urgency of the need for a solution [8]. Patient safety incidents due to acts of commission, excluding acts of omission, arise in 1–24 of every 100 primary care consultations, mainly due to misdiagnoses or medicines mismanagement; up to 44% of these lead to severe harm [9]. Both inappropriate- and under-prescribing increase mortality rates and hospital admissions [10]. Up to 92% of ADEs, adverse drug reactions (ADRs), and medicines' mismanagement (including patient safety incidents where patients and professionals contribute) are preventable [11–14]; more are due to poor monitoring than poor prescribing [15–19] and are dose-related [20,21]. Enhanced patient monitoring and/or reviewing would enhance efforts to ameliorate, if not resolve, the situation [9,16,17,19,22].

In the UK, 2008–11, 49% (of 7359) of older people (>64) took >4 medicines, up from 12% (of 7,614) in 1991–5, whilst the proportion taking zero medicines fell from 20% to 8% [23] and emergency admissions for adverse effects of medicine rose [24]. Life expectancy is no longer rising [25]. Regulation of primary care by incentivisation *via* the Quality and Outcomes Framework (QOF) did little to improve patient outcomes, despite considerable investment [26].

There is little evidence that existing interventions to improve the appropriateness of polypharmacy in older people improve outcomes [27]. Pharmacist-led innovations in pharmaceutical care and computerised decision support improve prescribing but not outcomes [27–32], and single-problem initiatives may be ineffective [33]. However, nurses' and carers' contributions to medicines optimisation remain unexplored.

Comprehensive, systematic multi-professional approaches are needed to ameliorate iatrogenic harm from ADRs [2–4]. In trials and observation studies, the ADRe Profiles for mental health medicines and respiratory medicines improved the processes and outcomes of care and medicines use for care home residents [4–6], and outpatients [7]. ADRe for polypharmacy (ADRe-p) was developed because we knew nurses' and patients' concerns about ADRs from mental health medicines were seldom recognised or communicated to prescribers unless ADRe was used [2,3]. In this study we sought to 'spread' the intervention to another geographical region, and to people prescribed multiple medicines [34].

## Aims

We aimed to investigate how deployment of ADRe-p changed: 1) residents' clinical wellbeing, measured as the number and nature of patients' problems and care quality as recorded on ADRe-p and in care records; 2) prescription regimens; 3) medicines optimisation; and 4) healthcare resource use. We also explored the aetiologies of problems identified and stakeholders' perspectives of the ADRe-p initiative.

## Methods

### Study design

This mixed-method process evaluation integrated and explored data from residents' ADRe-ps, medicines administration record (MAR) charts and care home records, at the start of the study and 5–10 weeks later, plus semi-structured interviews with healthcare professionals and service users. We followed SQUIRE (Standards for Quality Improvement Reporting Excellence) 2.0 [35] (S1 File).

## Setting

Between December 2018 and May 2019, ADRe-p was introduced into three independent private sector registered care homes with 158 residents in one Welsh University Health Board (UHB), unconnected with previous research sites [4,6]. Homes used electronic or paper records: all MAR charts were on paper. Prescriptions were issued by GPs and nurse prescribers, working with specialist teams e.g. community mental health, Parkinson's disease. Medicines were dispensed by community pharmacists. These included preparations for minor ailments, which are normally purchased by ambulatory service users. Patient records are regularly inspected to ensure they meet the standards of the Care Inspectorate Wales (CIW) https://careinspectorate.wales/about-us. Inclusion criteria for care homes were:

- Providing residential or nursing care or both to service users meeting inclusion criteria below.

- Willing to use the ADRe-p Profile in routine practice

- Staff aware of the Mental Capacity Act 2005, and willing to take informed consent.

    Residents' inclusion criteria were:

- Resident at the care home and expected to continue to live there for 1 year

- Currently taking >3 prescribed medicines daily. This definition of polypharmacy is used by Cochrane reviewers [27]. We were unable to adopt definitions based on appropriateness or duration of prescriptions before gaining consent to view records.

- Willing and able to give informed, signed consent themselves, or where capacity was lacking, a consultee was willing to give advice and assent to the resident participating.

    Exclusion criteria were:

- age <18

- Receiving active palliative care

- Not well enough to participate, as appraised by their nurses

## Our intervention: ADRe-p

ADRe-p engages nurses and carers in the multidisciplinary discussion around medicines management [2–4]. Searches indicate there is no alternative comprehensive, systematic patient assessment of problems potentially related to prescribed medicines [3,16,36,37]. ADRe-p is intended to support care home staff to raise concerns about possible ADRs or ADEs they have noticed. Supporting information provides possible explanations for observed signs or symptoms (Fig 1).

   **Developing ADRe-p.**   ADRe was adapted to encompass medicines commonly prescribed in primary care, including medicines for cardiovascular, respiratory conditions and diabetes, by addition of seven symptom-related questions, and removal of three. Decisions were based on an empirical review of the England and Wales patient safety incident reports describing signs and symptoms of adverse drug reactions [8], and review by our expert advisory group, including pharmacists, nurses and general practitioners. All ADRe-p items were rated anonymously by three pharmacists and two nurses. All items were rated important and 93/102 as of the highest importance to delivery or outcomes of care. (Changes from ADRe to ADRe-p are

listed in Table A in S2 File). ADRe-p was cognitively tested with two service users and one care home employee to ensure understanding. Sixteen suggestions were made: seven were adopted on ADRe-p, and nine were accommodated in the supporting information for carers and nurses.

## Approach and recruitment

Following ethical approval, February 2018, all 27 care homes in one district of the UHB were contacted by their pharmacist, and agreed to telephone contact by the researcher. When interest was expressed, researchers visited and provisionally arranged for adoption of ADRe-p, whilst the sponsor (the UHB) prepared research contracts for the universities and care homes. When the 171 page contracts (including a 150 page protocol) for care homes were ready, December 2018, the researcher revisited to confirm each home's participation. The homes reviewed their contracts, completed their details and passed them to the sponsor. They then received a final version for signature. Participating nurse leads were briefed by researchers, who joined team meetings to discuss implementation. Residents and service users were screened for eligibility and recruited by their nurses (see Ethics). Care home staff were invited to participate in serial interviews, and service users, pharmacists and GPs to single *post hoc* interviews.

## Outcome measures

1. Changes in signs and symptoms possibly related to adverse effects of prescribed medicines, as recorded on ADRe-p, using residents' notes as supplementary information.

2. Prescription changes in the same timeframe, as recorded on MAR charts.

3. Multidisciplinary medicines optimisation.

4. Costs of administering ADRe-p and changes in resource use resulting from ADRe-p administration. Costs were based on 2018/19 price year and taken from the Personal Social Services Research Unit (PSSRU) Unit Costs of Health and Social Care [38].

## Sample size

We had planned a larger study, but were unable to recruit. We had hoped to introduce medicines' monitoring into seven care homes, each with 26 residents prescribed >3 medicines (estimated 90% residents). Previously, with participants acting as their own controls, de-prescribing occurred in 8.5% more participants (12.1% vs. 3.6%) when ADRe Profiles for mental health medicines were used. We hypothesised that ADRe-p would be twice as effective as its predecessor for mental health medicines. Based on the reported intra-cluster coefficient (0.02) [6], 182 participants were needed to detect this difference with 80% power and 5% significance [39], allowing for 5% loss to follow up. The sample size is not based on the primary outcome (clinical gain) because all participants in the 2015 trial had several clinical gains recorded [6].

## Data collection

The care home records were examined before and after the introduction of ADRe-p to identify any changes in health status and signs and symptoms of potential ADRs (HP, SJ). Serial ADRe-ps administered by care home staff (at least two per resident), including pharmacists' comments, and MAR charts were copied, and collected by researchers. Problems and lists of

prescriptions were transcribed, and changes between the first and final ADRe-p were noted. HP and SJ collected and analysed data.

## Analysis

Data were entered into excel via the electronic version of ADRe-p, imported into SPSS version 26 [40], checked and described for first and last ADRe-p profile administrations. To explore the recorded changes and potential for further clinical gains, putative aetiologies of residents' signs and symptoms were ascribed, using clinical information from care home records, including, but not restricted to, information on ADRe-p, and MAR charts, and checked by all authors using manufacturers' literature and formularies. Juxtaposition of ADRe-ps and MAR charts generated suggestions for changes to improve the processes and possibly the outcomes of care. The prevalence of problems on the first and last profiles was compared with two by two contingency tables, using McNemar's test for related dichotomous variables [41].

All interviews were uploaded *via* digital media for transcription using a standard operating procedure (SOP) to ensure confidentiality. Interviews were transcribed *verbatim* and anonymised. Transcripts were thematically analysed alongside the data of all 19 completed cases, to identify, analyse and report patterns within data [42]. Two researchers (HP, SJ) ensured validity of the analysis and verified interpretation. Disagreements were resolved through discussion. Interviews and case reports were integrated with outcome measures documented on ADRe-p to enhance understanding and validity, and contextualise the ADRe-p data [43,44].

## Ethics

The local NHS Research Ethics Committee approved the study on 19th December 2017, and gave final clearance on 27th February 2018 (reference number 17/WA/0391, IRAS ID 237933). The Research and Development (R&D) department of the sponsoring UHB approved project start on 18th September 2018. Honoraria of £400 were paid to participating homes to cover costs, including staff time expended recruiting participants, liaising with researchers, and using facilities such as copiers.

Written and verbal information was offered to all potential participants. Written informed consent was obtained for all data collection, including completion of ADRe-p within routine care. Residents meeting inclusion criteria were approached by both registered nurses and carers and asked to consent to researchers reviewing their clinical records, including completed ADRe-p Profiles and MAR charts. Signed consent was taken by staff fully aware of the Mental Capacity Act 2005. Where nurses judged that residents lacked capacity to consent, consultees signed on their behalf; consultees were relatives or professionals not involved in the study, who were in regular contact with the participants [5,6]. The study was not unduly invasive, in that it did not expose participants to risks beyond the experiences of daily life or routine medical examination [45]. All previous participants have benefitted [4–7,46]. All interviewees were competent adults, and gave signed consent to audio-recording.

## Results

### Recruitment and retention

Of the 27 homes telephoned, three signed the contract and participated. Ten initially agreed to participate. Seven dropped out due to delays and complexities of the contracting process or changes in circumstances or staffing (Fig 2a). Of the 158 residents, 90 were ineligible and 23 were not approached. 16 families found the consenting process too difficult. 26 residents were recruited, and 19 completed at least two ADRE-ps on separate occasions; two passed on, three

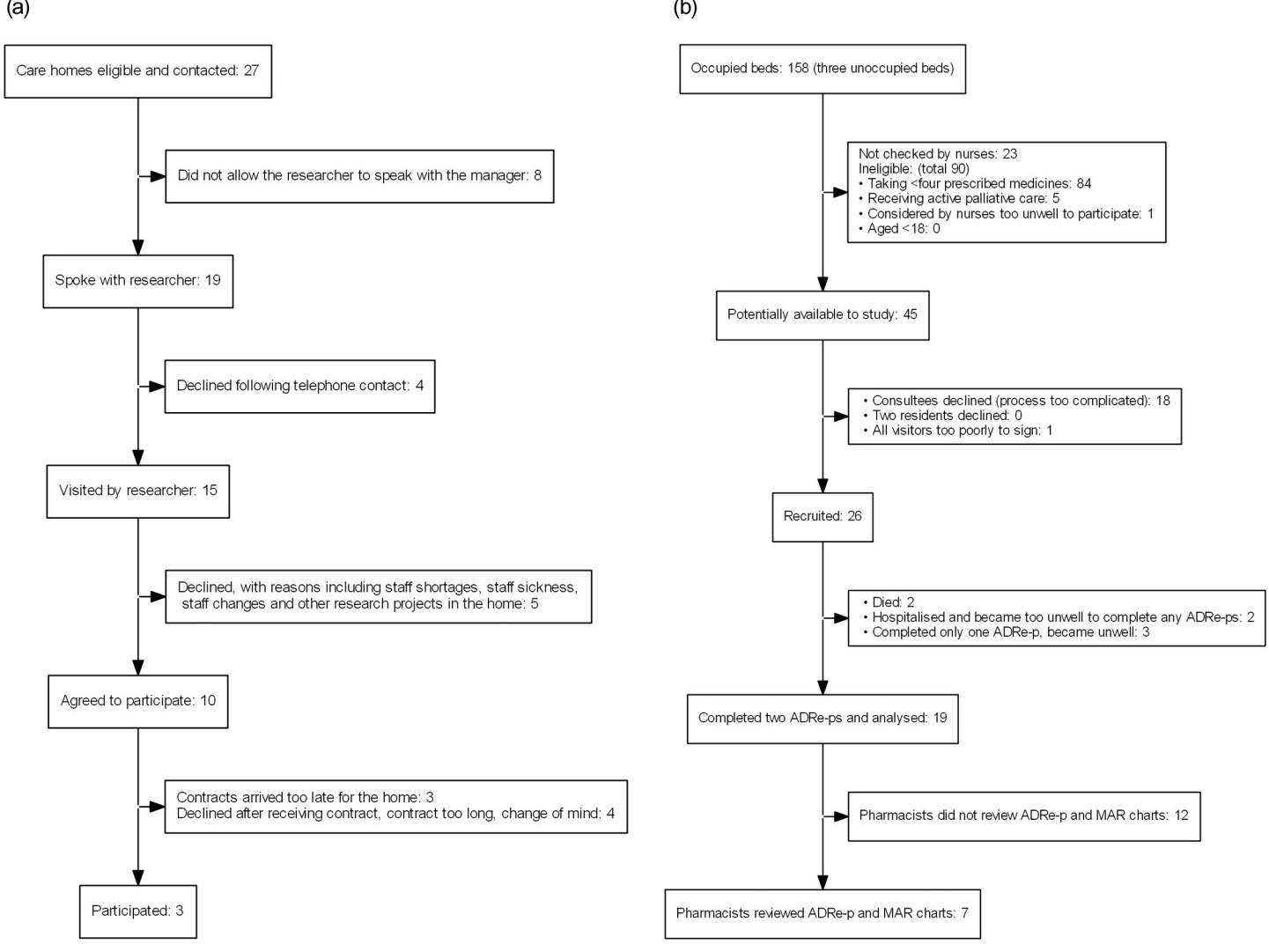

**Fig 2.** a. Care home recruitment: Flow diagram. b. Resident recruitment: Flow diagram.

became too unwell, and two were hospitalised (including a frail lady of 94 with acute kidney injury prescribed lithium).

Four of the five pharmacists working with the homes reviewed ADRe-p for 7 residents [5 (H2) 2 (H3)], within the study's timeframe. (Fig 2b) Of 28 interviews planned, 12 were undertaken and transcribed: service users (2), care home nurses or carers (6 –one twice), carers (2), pharmacists (3). All GPs declined interviews, asking pharmacists to convey their views, since they felt that GPs were not directly involved in the study.

**Participants.**  Researchers' initial impressions of the care homes were spotless floors, engagement with the community, and acts of kindness from staff. Medicines were invisible, but were affecting people's lives. By reading residents' notes (some 20 electronic files or paper documents per resident) we were able to glean 10–25% of the information needed to complete ADRe-p: this took researchers (SJ, HP) ~1 hour per resident per review. For example, seizures, insomnia, postural hypotension and extra-pyramidal symptoms (EPS e.g. Parkinsonian

**Table 1. Summary of outcomes for residents (n = 19).**

| Observations Numbers of: | Mean [SD] | Median [25th to 75th centile] | Range (min-max) |
|---|---|---|---|
| Problems identified/resident 1st profile | 15.84 [4.41] | 16 [14–19] | 19 [6–25] |
| Problems identified/resident last profile | 14.05 [4] | 13 [11.5–16] | 14 [8–21] |
| Changes to care by nurses/resident | 2.93 [1.7] | 3 [2–3.5] | 6 [1–7] |
| Positive outcomes of care (clinical gains)/resident | 3 [1.67] | 3 [2–4] | 7 [0–7] |
| Suggestions/resident from researchers | 5.11 [1.76] | 5 [4–6] | 8 [2–10] |
| Medicines prescribed 1st profile | **15.11** [4.1] | **14** [12–17] | 14 [10–24] |
| Medicines prescribed 2nd profile | **14.95** [4.65] | 13 [11.5–19.5] | **17** [7–24] |

SD standard deviation.

movements for residents prescribed antipsychotics) were only recorded on ADRe-p. ADRe-p was completed by carers, and copies were passed to registered nurses, pharmacists and prescribers.

The 19 residents completing two ADRe-ps were aged 35–92 (mean 74.8 [15.2]); 12 were female. Six were prescribed antipsychotics, eight AEDs, 14 antidepressants (3 prescribed >1 antidepressant), five sedatives, 10 medicines for dementia (3 prescribed >1 medicines in this class); only one resident (2.7) was prescribed no mental health medicines (Table 1).

## Outcomes

**1. Clinical gains and relation to intervention.** A mean [SD] of 2.93 [1.71] changes were made to the process of care per resident (range 0–6), including referrals, investigations and prescription changes (Table 1). For example: nurses were prompted to directly request GP reviews of medicines (6), and signs and symptoms, including extra-pyramidal symptoms (EPS), tremors, sleep problems, dehydration, and pain were recorded for the first time. Clinical gains were identified for 17/19 residents (mean 3 [1.67], range 0–7) (Table 1). Examples include: 6 residents no longer in pain, 3 no longer experienced convulsions, 1 had dyspnoea treated, laxative prescribing was adjusted to reduce diarrhoea for 2, falls ceased for 2 (of 4 noted as falling and of 5 able to stand). As carers became more familiar with some problems, such as EPS, they were more likely to record them (Tables B & C in S2 File).

Fewer residents experienced pain and confusion, and some residents became free of diarrhoea (4), asthenia (4), insomnia (4), aggression (3), swallowing difficulties (2), falls (2) and seizures (3). Few new problems arose, and no clinical deterioration was noted. However, not all problems were solved for all residents (Table 2).

The improvements in outcomes were diverse, and some (e.g. xerostomia, constipation, restlessness) were recorded only for a single resident (Tables B & C in S2 File). (Gains are listed in Table 3). The carers explained how they achieved these improvements, for example:

"*ADRe-p is a prompt to go further with the GPs, and obviously give nurses a better understanding and question: do you think this could be from medication? A resident (1.3) had a rash, now we said: they've got a rash do you think it could be from medication? So you automatically look a bit deeper than you would've before. (. . .) One resident (1.3) had an all-over rash, it was quite aggressive and then obviously when the GP came we were saying: she's on this medication and it's prompted us to go through this because of the ADRe-p, and then they worked with us on obviously identifying what the rash could be and giving the correct*

**Table 2. ADRe-p Profile items and responses: Observations and questions.**

| Item on ADRe-p profile | | | Problem on final profile | | P* | Unadjusted OR | 95% CI |
|---|---|---|---|---|---|---|---|
| | | | No | Yes | | | |
| Bowel control | Problem on 1st profile | No | 14 | 1 | 0.38 | ** | |
| | | Yes | 4 | 0 | | | |
| Low energy | Problem on 1st profile | No | 9 | 1 | 0.38 | 11.25 | 0.97–130.22 |
| | | Yes | 4 | 5 | | | |
| Sleep problems | Problem on 1st profile | No | 12 | 0 | 0.13 | ** | |
| | | Yes | 4 | 3 | | | |
| Aggression† | Problem on 1st profile | No | 12 | 0 | 0.25 | ** | |
| | | Yes | 3 | 3 | | | |
| Confusion | Problem on 1st profile | No | 7 | 1 | 0.38 | 12.25 | 1.08–138.98‡ |
| | | Yes | 4 | 7 | | | |
| Pain | Problem on 1st profile | No | 6 | 0 | 0.03‡ | ** | |
| | | Yes | 6 | 7 | | | |
| Swallowing difficulties | Problem on 1st profile | No | 14 | 0 | 0.25 | ** | |
| | | Yes | 2 | 3 | | | |
| Falls | Problem on 1st profile | No | 15 | 0 | 0.50 | ** | |
| | | Yes | 2 | 2 | | | |
| Seizures | Problem on 1st profile | No | 16 | 0 | 0.25 | ** | |
| | | Yes | 3 | 0 | | | |

*McNemar's test for related dichotomous variables, assesses change in response in both directions.

**Odds ratios could not be calculated due to zero in cells. This happened where the problems were addressed by the final profile.

†Missing data detailed in Tables B & C in S2 File. Cases with any missing value removed.

‡statistically significant.

*medication to address the rash." (. . .) the study has helped us identify little bits to push forward."*

(*Carer, H1*).

This sentiment was endorsed by pharmacists. The changes in medicines and problems are recorded in Table C in S2 File.

Not all residents' problems were addressed by the time of final record review (Table 2). Researchers made two to ten further suggestions per resident (mean 5.11 [1.76], Table 1). For example: three residents with hypotension (SBP<110mmHg) were prescribed antihypertensives (1.1, 1.2, 3.8), and, of six residents prescribed antipsychotics for >6–8 weeks, five had evidence of extra-pyramidal symptoms (EPS) (one also experienced insomnia when risperidone was given *nocte*). Two residents (both prescribed risperidone) had no improvements in outcomes, but we noted how some problems might have been addressed, for example, by adjusting analgesia or administering risperidone *mane* (Table 4 gives details).

**2. Prescription changes.** At least one medicine was discontinued for seven of 19 residents, and doses reduced for a further five. For example, zopiclone was de-prescribed for two, and senna and paracetamol were changed to PRN (*pro re nata*) (one each). Medicines were added or increased for nine residents. For example, two had skin conditions treated, and one (2.2) a salbutamol inhaler prescribed, alleviating dyspnoea. Overall, prescribing was reduced from

**Table 3. Clinical gains for residents: Comparison of first with final ADRe-p profiles (n = 19).**

| Resident | | | |
|---|---|---|---|
| ID | Age | | Clinical changes |
| 1.1 | 85–89 | F | **Outcomes of care**<br>1. **Pain** no longer present: paracetamol now given when needed.<br>2. **Heartburn and anorexia** resolved when iron discontinued.<br>3. **Confusion and sedation** appear to have resolved, possibly because appetite has recovered and heartburn is no longer troublesome at night.<br>**Process of care**<br>1. Gabapentin stopped when study was announced, before data collection: this likely ameliorated confusion. |
| 1.2 | 60–64 | M | **Outcomes of care**<br>1. **Pain** no longer present. This is probably due to regular analgesia, rather than relying on PRN requests from an inarticulate resident. However, the extra codeine may be causing sedation.<br>2. **Seizures** no longer reported, possibly due to improved pain management.<br>3. **Swallowing** difficulties resolved.<br>**Process of care**<br>1. Dietician review.<br>2. 'Flu immunisations completed.<br>3. ADRe-p was the only record of vital signs. |
| 1.3 | 65–69 | F | **Outcomes of care**<br>1. **The rash** has resolved, probably due to the newly prescribed Piriton™.<br>2. **Seizures and**<br>3. **aggression** resolved. AEDs were being given same time as bulk laxative. Therefore, AEDs started to work when bulk laxative stopped.<br>4. **Diarrhoea & double incontinence** ceased, likely due to discontinuation of bulk laxative.<br>5. **Swallowing difficulties and**<br>6. **Painful gums** have resolved, likely following assessments triggered by ADRe-p.<br>**Process of care no change** |
| 1.4 | 70–74 | F | **Outcomes of care**<br>1. **Hypotension and**<br>2. **Incontinence** appear to have resolved. This may be due to the reduced bioavailability of generic galantamine (there has been a generic substitution for Luventa®).<br>3. Swallowing difficulties have resolved, but not clear if this is related to ADRe-p.<br>**Process of care**<br>1. **Scalp itching** is now being treated.<br>2. Vitamin D now prescribed for fracture prevention/general health. |
| 1.5 | 80–84 | F | **Outcomes of care**<br>1. **Restlessness, aggression,**<br>2. **Pain, constipation, broken skin & missed meals** were not recorded by profile 3.<br>3. Regular (not missed) meals probably corrected hypoglycaemia, thereby improving agitation and aggression. Likely, ADRe-p drew attention to the missed meals.<br>**Process of care**<br>1. Extra-pyramidal symptoms (EPS), nausea and dehydration were recorded for the 1st time. EPS were probably due to risperidone, which has been prescribed for >8 weeks: delayed onset of EPS, particularly tardive dyskinesia (jerky movements now seen) is usual.<br>2. OT referral: OT requests behaviour support, but doctor says this is unhelpful. |
| 1.6 | >90 | F | **Outcomes of care**<br>1. **Weakness and**<br>2. **missed meals** have resolved. Changing zopiclone to PRN in follow up notes probably helped by alleviating weakness.<br>**Process of care**<br>1. Doctor asked to review cough—says observe only.<br>2. Dysphagia screening arranged: advice offered.<br>3. Medication error (no details found).<br>4. EPS were being recorded by end of study. Onset may be delayed: a likely cause is a high dose of duloxetine for an underweight elderly lady.<br>5. No other records of oxygen saturation. |

(*Continued*)

**Table 3.** (Continued)

| Resident | | | |
|---|---|---|---|
| **ID** | **Age** | | **Clinical changes** |
| 2.1a | 35–39 | F | **Outcomes of care**<br>1. Pain resolved by study end. Paracetamol & ibuprofen administration appear to be less frequent.<br>2. Behaviour appears to have eased, but anxiety remains.<br>3. Seizures appear controlled.<br>Cognitive decline may have stabilised.<br>**Process of care**<br>1. Resident prescribed carbamazepine and immobile. Accordingly, pharmacist requested vitamin D intake to be discussed with doctors, but no action by study end. (Potential fracture prevention.) |
| 2.2a | 70–74 | F | **Outcomes of care**<br>1. Dyspnoea has resolved now that asthma is treated.<br>2. The resident is no longer in pain. This may be due to the reduction of rivaroxaban (indication without aspirin not clear) or the decision to change the suprapubic catheter q.4 weeks, not q. 8–12.<br>3. Suprapubic catheter no longer blocking.<br>4. Rash has resolved: cause uncertain but likely the reduced dose of anticoagulant.<br>5. FBC reported as normal. Iron discontinued.<br>**Process of care**<br>1. ADRe-p was the only record/documentation of oedema or tremors.<br>2. A new eye-care preparation has been prescribed. |
| 2.3a | 65–69 | M | **Outcomes of care**<br>1. Seizures reported pre-study are no longer present. A consultant letter requested documentation of seizures. 6 months later, after start of study, a record of seizures has been introduced & 2.3a is now free of seizures. Macrogols (Laxido®) are given less often: it appears they were co-administered with AEDs, reducing effectiveness of AEDs.<br>2. Lost 5kg weight, BMI reduced from 33 to 31.5.<br>3. Diarrhoea no longer present, likely due to reduced laxative administration.<br>4. Frustration no longer reported, uncertain whether ADRe-p, improved seizure control, dental extraction, end of diarrhoea have helped with this.<br>Swallowing difficulties no longer reported.<br>**Process of care**<br>1. Dental extraction, but pain persists.<br>2. Risk of dehydration reduced, possibly due to decreased bulk laxatives. |
| 2.4a | 40–44 | F | **Outcomes of care**<br>1. Diarrhoea resolved now laxatives only PRN, following pharmacist's advice.<br>2. BP,<br>3. BMI and<br>4. Confusion have improved now pregabalin dose reduced.<br>**Process of care**<br>1. ADRe-p is the only record of vital signs. |
| 2.6b | 80–84 | M | **Outcomes of care**<br>1. Falls no longer a problem. This may be attributed to the medication changes and pharmacist review, particularly reduced dose of antipsychotic & reduced anticholinergic burden as promethazine discontinued plus zopiclone discontinued. Aripiprazole is noted as being sedative & causing insomnia, and making administration *nocte* might have helped.<br>2. BMI improving and not regarded as a problem now that nutrition supplement is prescribed, aripiprazole reduced & anorexiant discontinued.<br>**Process of care** no change |
| 2.7d | >90 | F | **Outcomes of care**<br>1. Sleep,<br>2. balance and falls have improved, possibly due to regular administration of nutrition supplements.<br>**Process of care**<br>1. Possible hypothyroidism explored by pharmacist.<br>2. Medicines are no longer missed, which may account for the appearance of agitation or indigestion. |

(*Continued*)

**Table 3.** (Continued)

| Resident | | | Clinical changes |
|---|---|---|---|
| ID | Age | | |
| 2.8b | 80–84 | F | **Outcomes of care**<br>1. Gait appears to be improved now the opioid and hypnotic have been discontinued following pharmacist review.<br>2. Convulsions and panic attacks are no longer reported, and this may be due to discontinuation of buprenorphine (a rare ADR).<br>3. Hypertension appears to have resolved, and this may be related to discontinuation of paracetamol.<br>**Process of care** no change |
| 3.2 | 80–84 | M | **Outcomes of care no change** (only 6 problems listed)<br>**Process of care**<br>1. Dental problems identified.<br>2. Codeine increased to treat pain, but ineffective.<br>3. FBC indicating anaemia reported, but not linked to poor healing. No actions.<br>4. Medicines administered regularly.<br>5. Tablets no longer crushed. |
| 3.3 | 80–84 | F | **Outcomes of care**<br>1. Pain now confined to 1 shoulder.<br>2. Benefits of bringing prescribing of dopamine agonists into line with guidelines are yet to be fully realised.<br>3. Insomnia no longer a problem.<br>4. Incontinence appears to have resolved.<br>**Process of care**<br>1. ADRe-p provided the only record of posture and movement disorders. This has triggered medicines review. After Profile 1, the nurse asked the GP to review medicines. The GP contacted the Parkinson's nurse and the mental health nurse to do these reviews. The home was requested to monitor resident, but there were no specifications.<br>2. Dopamine agonists were increased to address tremors and the gap in administration has been closed from 16 to 8 hours (sup info p.9): the only evidence of effectiveness is pain reduction.<br>3. Falls risk reassessed by study end.<br>4. GP & mental health nurse disagreed as to whether to use lorazepam PRN or quetiapine. The care home nurses became actively engaged in the decisions, and consulted by prescribers.<br>5. Need for calcium and vitamin D supplements highlighted by pharmacist. |
| 3.6 | 75–79 | M | **Outcomes of care**<br>1. Symptoms of anxiety and tiredness have abated, but aggression remains.<br>2. Dry mouth has improved, explaining why appetite has returned.<br>**Process of care**<br>1. GP contacted and resident reassured regarding visual hallucinations (dopamine agonists prescribed).<br>2. Overuse of senna has been curtailed.<br>3. Dehydration risk is now recognised.<br>4. Missed doses are highlighted. |
| 3.7 | 75–79 | F | **Outcomes of care** no change<br>**Process of care**<br>1. Assessed with Abbey pain scale.<br>2. April 2019, nurses requested meds review. May 2019, GP reports no EPS, and risperidone continued.<br>3. Unresponsive episodes reported for the first time: no MDT engagement by study end. |

(*Continued*)

**Table 3.** (Continued)

| Resident | | | |
|---|---|---|---|
| **ID** | **Age** | | **Clinical changes** |
| 3.8 | 85–89 | F | **Outcomes of care**<br>1. Restlessness,<br>2. hallucinations, panic attacks and aggression have improved, probably due to medication review requested.<br>3. Xerostomia has improved, possibly due to prescription changes.<br>**Process of care**<br>1. GP asked by nurses to review meds at start of study: mirtazapine dose split, diazepam changed from regular to PRN.<br>2. April 2019, GP asked to review edoxaban for bruising & nosebleeds, but continued anticoagulant. Again asked to review May 2019 and sertraline increased to 100mg. Different GP each time. ADRe was the only report of blood loss identified.<br>3. Dehydration risk recognised and addressed. |
| 3.10 | 80–84 | M | **Outcomes of care**<br>1. Pain has improved.<br>2. Constipation and bowel control have improved, likely due to deprescribing co-codamol, Laxido® and enema.<br>3. Oxygen saturation is marginally improved, which might be attributable to stopping zopiclone.<br>**Process of care**<br>1. Co-codamol has been discontinued following medicines review.<br>2. Active review of bowel control: Laxido was noted as a cause of diarrhoea, and this has been stopped.<br>3. Fenbid® gel stopped. |

Home 2 was divided into units specializing in caring for people with dementia (2b, 2d) and adults with physical disabilities (2a).

mean 15.1 [4.1] to 14.9 [4.7] items/resident, including emollients and nutritional supplements (Table 1). Examples of changes are listed in supplementary file Table C.

Prescription changes were brought about by: nurses requesting GP reviews (six residents: 2.6, 2.8, 3.6, 3.7, 3.8, 3.0); pharmacists actively engaging with ADRe-p (Fig 1); routine contact with GPs and non-medical prescribers (1.4, 1.6, 3.2, 3.3); nurses managing symptoms via existing PRN orders (1.6, 2.4, 3.8) (Table 3)

"*Helps us identify reactions to medication, which prompted us to go to the GPs, have a better understanding, and being able to challenge the GP.*"

[*H1, N2*].

However, most medicines were unchanged, but not necessarily unchallenged, during the 3 months of the study, and repeat prescriptions were the norm (Table C in S2 File has examples). A concerned service user related to her mother's care in another home:

"*Her head down. . . they tell me it's the side effects of antipsychotics (. . .) this* [study] *is a good thing, should be put in place for people, everybody (. . .) they just keep repeat prescriptions.*"

(*SU1, H2*).

**Table 4. Residents with no recorded clinical gains.**

**Resident 3.2** (male, aged 80–84) **Pain remains**

| Problems ADRe-p profile 1 | Possible causes |
| --- | --- |
| BMI 35 | Risperidone |
| Broken skin/poor healing (leg ulcers) | Worsening heart failure, obesity, risperidone |
| Pain | Risperidone, statin, possibly gout from furosemide. |
| Vision problems (cataract) | Risperidone |
| Urinary retention | Risperidone, diuretic, rivastigmine, and/or opioids |
| Long term urinary catheter | |
| Tablets crushed—now resolved | |
| Medicines not taken at same time each day—now resolved. | |

**Medicines first and final review**–no changes

- Apixaban 2.5mg tabs, take 1 2x a day (morning & teatime 5pm)
- Bisoprolol 1.25mg tabs, take 1 each day
- Codeine Phosphate 15mg tabs, take 1–2 up to 4 times a day when required? Given
- Epaderm® cream, apply daily
- Evacal® D3 chewable tab, take 1 2x a day (morning & teatime 5pm)
- Furosemide 40mg tabs, take 1 in the morning
- Laxido® Orange oral powder, take 1 a day in water 8 given
- Omeprazole 40mg gastro-res cap, take 1 daily
- Optiflo® S 0.9% sterile soln, not given
- Paracetamol 500mg tabs, take 2 every 4–6hrs when required. Given frequently.
- Risperidone 500mcg tabs, take half in the evening
- Rivastigmine 3mg caps, take 1 2x a day (morning & teatime 5pm)

Simvastatin 40mg tabs, take 1 in the evening

**Pharmacist's recommendations:** FBC shows HB at 119, slightly low

**Researchers' suggestions for discussion:**

1. Pain remains: it is not responding to codeine, which is likely causing constipation, and is not recommended for people with dementia.
   Has gout been considered (furosemide is prescribed)?
   Risperidone can cause joint disorders and stiffness and, therefore, pain.

2. 2 antihypertensives are prescribed whilst BP is 116/67, 110/68mmHg on standing: consider postural hypotension.
   Beta-blockers are not recommended for people with dementia, but any reduction will need to be gradual.
   Furosemide likely essential.

3. Attempts to break a 500mcg risperidone tablet are unlikely to give even doses—liquid is available. Tablets are film-coated.

4. Risperidone is indicated for 6–8 weeks in the elderly with persistent aggression (not noted as present).
   (Interaction with furosemide and risk of CVA, see manufacturer's literature). It may be contributing to weight gain and worsening heart failure.

5. Consider risk of catheter blocking if using calcium (vitamin D alone is not a risk).

6. Urinary retention is reported: consider risperidone, rivastigmine, codeine as possible causes.

**Resident 3.7** (female, aged 75–79) **Behaviour Difficulties**

| Problems ADRe-p profile 1 | Possible causes |
| --- | --- |
| Temp 35.7 | Incomplete contact between thermometer and skin |
| BMI 28 | Cetirizine & mirtazapine can increase appetite |
| Unable to stand | Illness, memantine, and/or donepezil (dizziness) |
| Abnormal posture—not on last profile | Risperidone, cetirizine (rarely), sertraline (neuromuscular dysfunction), and/or memantine (balance) |
| Immobile | Illness, risperidone, memantine, and/or donepezil |
| Dementia/cognitive decline | Illness, cetirizine, and/or risperidone |
| Behaviour problems | Risperidone, cetirizine, sertraline, mirtazapine, and/or donepezil |

(*Continued*)

**Table 4.** (Continued)

| | |
|---|---|
| Physical violence | Mirtazapine and risperidone can exacerbate symptoms of psychosis. Risperidone dose received is uncertain, as a 500mcg tablet is halved |
| Aggression | Donepezil, mirtazapine (rare). Hypoxia, sertraline, risperidone (akathisia), and/or memantine (psychotic disorder) |
| Agitation/anxiety | Risperidone, mirtazapine, sertraline, donepezil, and/or cetirizine |
| Confusion | Cetirizine, mirtazapine, memantine, risperidone. Attributed to dementia |
| Mood fluctuations | Attributed to dementia. Sertraline, mirtazapine, risperidone, donepezil |
| Insomnia | Risperidone *mane*, donepezil, cetirizine, and/or mirtazapine |
| Vision problems (glasses) | Risperidone can blur vision |
| Incontinent | Donepezil, risperidone. Not clear if retention an element |
| Unresponsive episodes | Idiopathic, memantine, donepezil and other epileptogenic medicines |

**Medicines first and final review**–no changes

- Cetirizine 10mg tabs, take 1 daily
- Co-Codamol 30mg+500mg tabs, take 2 4x a day when required not used
- Conotrane® 0.1%+22% cream, not given apply to affected area
- Donepezil 10mg tabs, take 1 at night
- Duraphat® 5000 toothpaste, use 2x a day
- Hydromol® ointment, not given apply to skin or use as soap substitute
- Ibuprofen 5% gel, apply up to 3x a day
- Loperamide 2mg caps, take 2 & then 1 after each bowel movement as required not used
- Lorazepam 1mg tabs, take half each day when required not used
- Macrogol® compound oral powder, take 1 sachet in water each day when required 1 given
- Memantine 20mg tabs, take 1 at night
- Mirtazapine 30mg tabs, take 1 at night
- Risperidone 500mcg tabs, take half at night
- Sertraline 100mg tabs, take 2 each day

**Pharmacist's recommendations**: none found

**Researchers' suggestions for discussion:**

1. Risperidone reaches peak absorption 1–2 hours after ingestion, and causes insomnia in >1 in 10 recipients (manufacturer's literature), as here. Therefore, should be given *mane*. Use appears to have exceeded 6–8 weeks.
2. Use liquid risperidone to avoid splitting tablets and ensure even dosing, particularly in view of behaviour problems.
3. Indication for cetirizine (anti-muscarinic) not clear, as no skin problems or hay fever noted (in March); review might reduce the anti-cholinergic burden and associated confusion and behaviour problems.
4. Sertraline is maximum dose; given the possible association with aggression, consider benefit to harm balance.
5. There is no diagnosis of epilepsy, but the unresponsive episodes and behaviour problems indicate a need to consider this diagnosis, particularly as memantine, donepezil, anti-depressants and anti-psychotics can cause or predispose to seizures.
6. Both anti-dementia medicines are recommended as monotherapy. The clinical benefit of the combination might be considered: memantine may be worsening respiratory or cardiovascular deterioration; oxygen saturation is borderline and likely contributing to confusion.

Co-prescribing of loperamide and Macrogol® appears incongruous, but administration records appear appropriate.

Note to Table 4: Problems underlined were not present on last profile.

**3. Making it happen: Multidisciplinary medicines optimisation.** Much was achieved by modifications to nursing care, despite "*everybody being so busy and very uptight*" (N1 H2). For example: changing a supra-pubic catheter every 4 rather than 8 weeks (2.2) addressed the problems of frequent blockage and alleviated pain; recording EPS to inform GP of problems (3.3); or improving intake (five residents). Nurses followed their usual practice of contacting GPs with concerns.

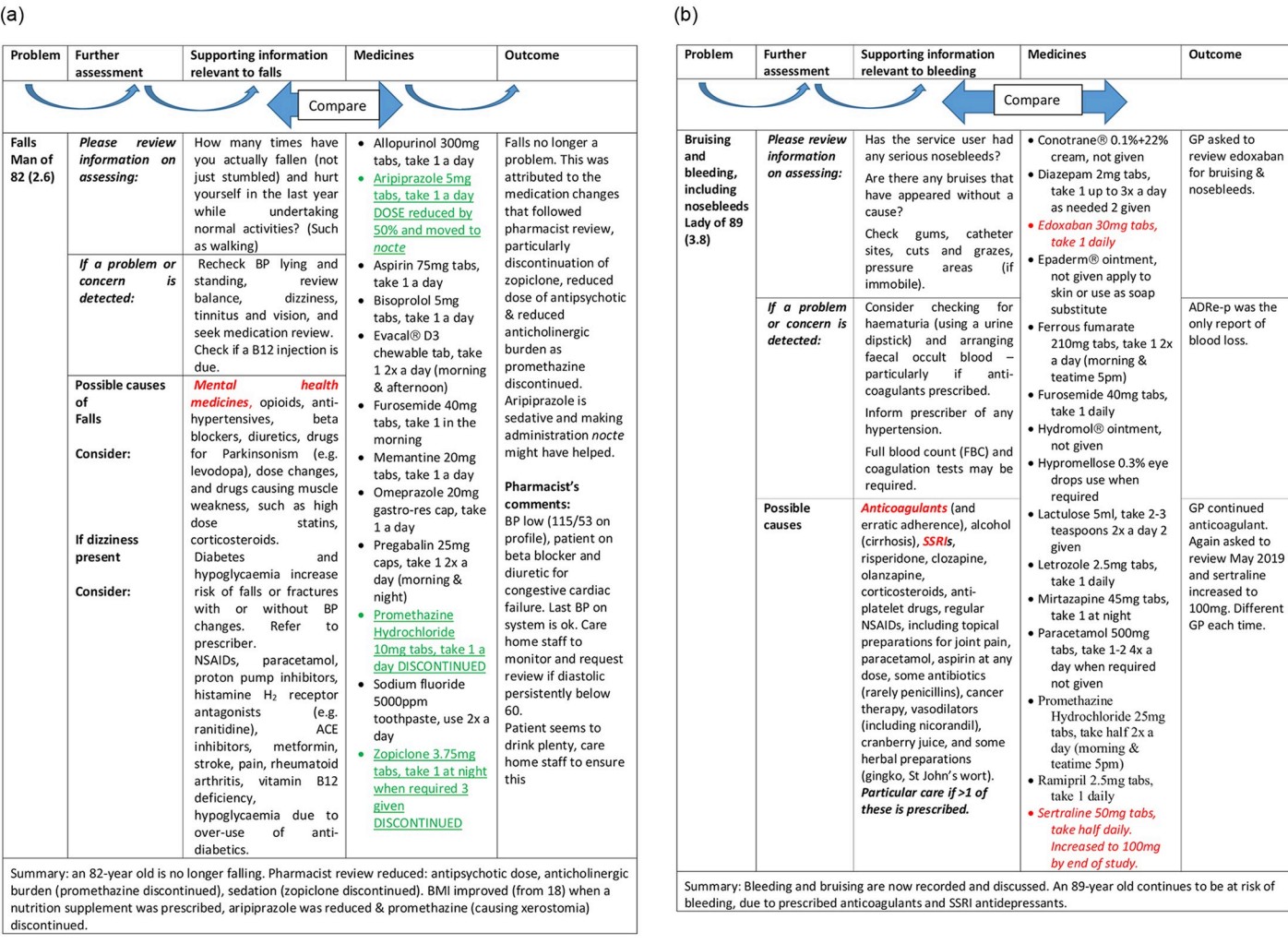

**Fig 3.** a. Linking problem to solutions, example 1: ADRe's information used—no more falls. b. ADRe's information not used—nosebleeds continue. Medicines underlined were discontinued or reduced. Medicines italicised were started or increased. Aetiologies emboldened were the most pertinent to the resident.

Where pharmacists engaged, residents benefitted e.g. by becoming free of falls (2.6, Fig 3a) or diarrhoea (Table 3, 2.4: 2 laxatives changed to PRN from regular administration). Pharmacists explained how pressures on the services militated against communication and GP involvement, and ADRe-p filled this hiatus in communications, particularly with non-verbal patients:

"*The GPs are busy; the nurses are busy, sometimes. The GPs head* over [to care homes] *when they can, at the end of their clinics. The nurses might've been stressed all morning, they are trying to explain themselves to the GP, the GP is trying to obtain a really good thorough history and it's difficult. Whereas, if you've got one of these* [ADRe-p], *you could just cross-reference, and be like "this is what I was concerned about, see". So yeah we were saying it is a really useful tool.*"

*P1, H2.*

Repeated use made ADRe-p easier and quicker "*5–15 minutes total*", N1, H3), and engendered a virtuous circle of learning, recognition and knowledge application:

"*. . . the more the nurses fill out these things, the more in their mind it's going to be, they'll know what the side effects are (. . .) they'll see something and automatically associate.*"

*P2, H2.*

"*. . .empower them [nurses] to say 'this has changed, and look I've got proof'". (. . .) "Like me, the GP might go in and see the patient for a snapshot of time, but the nurses are there with the patients a lot more than we are.*"

*P1, H2.*

However, not everyone fully engaged, largely due to pressure of work, and some problems went unattended. For example, when nurses asked the GP to review prescriptions of anticoagulants for resident 3.8 with nosebleeds and bruising, the GP added sertraline (indication not identified), which increases risks of haemorrhage (Fig 3b). Similarly, a resident with infections, a tracheostomy and suprapubic catheter was prescribed a high dose of pregabalin (450mg daily, presumably for pain or anxiety), which predisposes to infections (2.2). Where pharmacists did not engage, time pressures meant that complexities were sometimes overlooked. For example, a lady of 65 (1.3) with seizures was administered antiepileptic drugs (AEDs) simultaneously with bulk laxatives; when diarrhoea was noted on ADRe-p, laxatives were reduced and seizures ceased, possibly because the AED was then fully absorbed.

There were sometimes multiple prescribers. No formal communication systems between prescribers and carers and nurses giving daily care were observed. ADRe-p acted as shared documentation, and provided the only record of signs and symptoms indicating possible ADRs. ADRe-p encouraged nurses to seek medication reviews. For an 81-year-old with Parkinson's disease and antipsychotic prescriptions (3.3), the GP contacted the Parkinson's nurse and the mental health nurse to do these reviews. The home was asked to monitor the resident, but there were no specifications: ADRe-p filled this gap. The Parkinson's nurse increased dopamine agonists to address tremors and the overnight gap between administrations was narrowed from 16 to 8 hours (ADRe's supplementary information p.9). Pain decreased, as Parkinsonian stiffness resolved. There were discussions as to whether lorazepam as required (PRN) or quetiapine should be prescribed for behaviour problems. ADRe-p (and Supporting information) empowered the nurses and carers to join these discussions.

**4. Economic costs and benefits of ADRe-p.** We tested whether ADRe-p could be introduced to support existing routine care in homes, and demonstrated that many, but not all, ADRs are prevented when ADRe-p is used. Pharmacists confirmed that ADRe-p was addressing common problems

"*. . .umpteen blood pressure medicines, blood pressure in their boots, giddy and falling, and nobody ever questions it, whereas using these forms (. . .) will keep everything in an easy format (. . .) make the nurses think.*"

(*P1 H2*).

Administration of ADRe-p took nurses or carers 5–15 minutes and pharmacists 10 minutes per resident per month, introducing additional demands on staff time. However, this cost is minimal, and experience with ADRe-p leads to more efficient completion, keeping costs low. To estimate costs, we assume a Band 6 Community Nurse, at a rate of £46 per hour (including

salary and on-costs), undertakes all ADRe-p related activity. Costs include an initial review of resident taking ~30 minutes, (up to 1 hour if blood tests are included) (£23–46) plus 15 minutes per ADRe-p re-administration (£11.50). If ADRe-p is used monthly, it will cost £184 per resident in the first year and £138 in subsequent years. If ADRe-p is administered by a care home worker with an hourly salary of £23, as here, then administration costs fall to £92 in year one and £69 in subsequent years [38].

Use of ADRe-p may also lead to increased contacts with GPs. However, these additional time costs were again minimal. An additional GP contact would incur a cost of £39 per patient. Nurses discussed ADRe-p with GPs either at scheduled contacts or as a direct result of ADRe-p. Pharmacists took 10 minutes to review ADRe-p: costs of a Band 6 pharmacist are estimated at £46 per hour or £7.67 for a ten-minute review [38].

Health improvements reduced resource use: for example, three patients had changes in medicines that eliminated seizures, and two patients no longer had falls; both reduced seizures and falls will likely lead to fewer avoidable hospital admissions. A fall that results in an inpatient admission, perhaps for an intermediate hip procedure for trauma with no or minimal complications, would cost £3,284, plus additional costs for post-discharge care and rehabilitation. Even where there is no serious injury arising from a fall, attendance at A&E with a Category 2 Investigation with Category 1 Treatment (£155) costs slightly less than the annual administration cost of ADRe-p when completed by nurses, and is more than the annual cost if ADRe-p is completed by care staff. Through improved health and reduced falls, ADRe-p has significant potential to both improve patient health and lower health system costs overall. For more serious adverse events after a fall cost-savings would be even greater.

## Discussion

We aimed to identify changes following nurses', carers' and pharmacists' use of ADRe-p. Introduction of ADRe-p was followed by reduction in pain, seizures, falls, diarrhoea, dyspnoea and other symptoms for most residents, but more might have been done in some instances. Clinical gains were identified for 17/19 residents. Examples included: six residents no longer in pain, three seizure-free, one breathing easily, laxative prescribing adjusted to reduce diarrhoea for two, falls ceased for two (of four falling). ADRe-p collated patient information to minimise 'undesirable' signs and symptoms, optimise prescribing, and prevent medicines-related harm. It prompted multidisciplinary team engagement to address pain, usually by ameliorating stiffness or drawing attention to PRN regimens. However, there were instances of under-prescription of analgesia, as in other vulnerable populations [47]. Only careful review of medication records and patient history identified likely causes and candidate medicines for de-challenge (Fig 3a and 3b). However, sometimes, multidisciplinary teams felt too time-pressured (and stressed) to engage with the study, prescribing cascades [48], and the associated complexities presenting apparently insoluble problems [49].

### Bringing residents' and carers' voices into the multidisciplinary team

ADRe-p captures patients' perspectives and enhances patient-centred professionalism by direct questioning and profiling problems. Similar electronic patient self-report of selected symptoms is used by oncology trialists [50]. There are no short cuts to "asking the patient specific questions" [51] p.13, to generate shared documentation [52] and shared decision-making on medicines optimisation [53]. ADRe-p achieves this by initiating the dialogue needed to capture signs and symptoms. It also ensures patients receive information on ADRs, and helps professionals to attach the same importance to ADRs as patients do [54]: ADRe-p was welcomed by service users.

These findings echo those elsewhere: patients benefit when, together, professionals engage with the detail, complexity and totality of their problems. The initial time invested with patients leads to longer-term clinical gain and easier nursing [4–6,55]. Replication of findings in other settings and contexts strengthens the logical inferences to be drawn from geographically limited work [34,56]. Although human factors, leadership, organisational culture and multidisciplinary team-working vary between care homes and health boards [57], ADRe achieves similar outcomes.

## A difficult and orphaned task

Disengagement from medicines and their complexities remains a challenge to be addressed by medicines optimisation [58]. The GMC [59] p.9 advises doctors: "When you issue repeat prescriptions or prescribe with repeats, you should make sure that procedures are in place to monitor whether the medicine is still safe and necessary for the patient." (P.9, point 59). However, the Care Quality Commission [60] p.16 report that patients in care homes are not monitored because there are no systems to support the process, and professional responsibilities for this are not defined. This may, in part, be attributed to the uncertainty regarding nurses' roles in several aspects of medicines management [61], including engagement in monitoring, identifying and reporting potential ADRs [62].

De-prescribing in care homes is difficult, time-consuming, perceived as excessively high-risk [63] or thwarted by staff shortages [64]. Staff shortages were commonly cited as reasons for non-participation. However, in practice, completion of ADRe-p took carers or nurses 5–15 minutes and pharmacists 10 minutes per month. To succeed, despite organisational and professional barriers, discussions around de-prescribing need to be formalised and integrated into routine care [53]. Involving the staff spending most time with residents—nurses and carers— is crucial, and ADRe-p offers a feasible structure for enfranchisement and democratisation of medical knowledge [65].

Some 10% of healthcare is harmful, and 30% is wasted: identifying the 10% requires change [66], but change is more difficult where information is complex [49]. ADRs and prescribing cascades are hard to detect [47], and medical science remains unapplied [67]. Full evaluation of possible causes of residents' signs and symptoms requires close attention to the details contained in the 'undesirable effects' sections of SmPCs (Table 4, Table D in S2 File). Some residents are reportedly too poorly to realise any clinical gains [68], but ADRe's detailed symptom checking led to symptomatic improvement in our residents, when coupled with very dedicated nursing care (1.3, Table 4).

## New ideas in old structures

Demographic changes and increasing prescribing rates are likely to exacerbate ADR-related problems, but, currently, there is no defined way forward [69]. No alternative strategies to routinely or consistently check for ADRs have been suggested [69,70], ADR risk prediction tools are inadequate [71], and there is little evidence that electronic de-prescribing tools improve clinical outcomes [30]. Unfamiliar knowledge is not always welcomed, particularly by senior nurses and doctors [72,73], and an organisation's internally defined reward system determines which activities are pursued and which neglected [74]. To overcome the invisibility and disengagement with medicines, and 'kick start' the initiative, monitoring may need to feature in external incentivisation or regulatory systems. ADRe-p could help mediate such changes.

Although authorities consider monitoring patients an essential component of prescribing, no structures are offered [52,59,60]. ADRe-p offers a workable structure, and is needed to support medicines reviews [2,3,52,60,65,75–77]. The proposed changes in primary care services,

entailing weekly GP visits to care homes [78,79] will benefit from all information in a single shared document [52], rather than ~20 computer files.

## Limitations and strengths

This was a before-and-after observational study, without comparators. Our earlier randomised controlled trial demonstrated how ADRe alleviated pain, reduced prescribing of sedatives and increased nursing actions [6], but did not lead to ADRe-p adoption by policy makers. An observational approach was considered complementary [74], but we cannot assume *post hoc*, *ergo propter hoc*.

There are 34 definitions of ADRs, some contradictory, and most lacking inter-rater reliability [80]; where medicines were de-prescribed, re-challenge was considered an unacceptable risk. Accordingly, we make no claims regarding the aetiology of signs and symptoms recorded: ADRe-p only suggests that iatrogenic aetiologies be considered, rather than allow problems to be dismissed as inevitable consequences of age or disease [48,53]. Similarly, we were unable to adopt any of the 21 approaches to categorising harm emanating from healthcare-associated incidents [81], due to the uncertain duration of problems identified.

Limitations of single site work are reported [3,4]: this study was, in part, undertaken to address this by seeking replication in different geographical locations [56]. Neither we nor the care homes had access to GP records. Our assessments of the impact of ADRe-p relied on triangulation of completed ADRe-ps, MAR charts, care home notes (containing GP and consultant letters and referrals), fieldwork observations and interviews. Accordingly, our suggestions are presented as discussion for future work.

Our sample size was smaller than anticipated. This may have been attributable to delays (above), and a subsequent loss of resources for recruitment. We had originally anticipated support from an NHS-funded primary care nurse, and goodwill support from academic colleagues, but, due to delays in obtaining signatures on contracts, other commitments precluded participation when required. A minority of homes approached were recruited (3/27). This may have caused a volunteer bias, as elsewhere [4,82]: homes that declined often cited staffing issues, whereas those completing did not share these problems, and used zero or few agency staff. Similarly, of the five pharmacists working with the homes, four engaged in the study, and they were uniformly excellent.

The study was not designed to capture definitive data on resource use, but the evidence shows that ADRe-p can improve patient outcomes, and may reduce healthcare costs. ADRe-p involved existing staff, and therefore had relatively low delivery costs, increasing the cost-benefits of change.

We did not use quality-of-life measures, as we found that these fail to capture important symptoms, such as pain and emesis, are difficult to administer with people with cognitive impairment [6], and reviewers indicate that they are infrequently used in this area [28].

## Implications

ADRe-p relieved unnecessary suffering for participants. It was the only observed strategy for multidisciplinary team working around medicines optimisation. ADRe-p prevents ADRs becoming serious and improves care quality by: a) regular systematic checks and documentation of problems; b) transfer of information from care home staff to prescribers and pharmacists to optimise therapeutic regimens; c) recording change; d) involving the professionals closest to residents in decision around medicines, doses and timings, and providing them with supporting information. As the care home sector emerges from the COVID-19

pandemic, ADRe-p should form a crucial aspect of upskilling and enhanced multidisciplinary communication.

## Supporting information

**S1 File. SQUIRE2.0 revised standards for Quality Improvement Reporting Excellence plus.** (DOCX)

**S2 File Supplementary tables: Table A Changes made from ADRe to ADRe-p Table B ADRe-p Profile items and responses: Vital signs (n = 19) Table C ADRe-p Profile items and responses: Observations and questions (n = 19) Table D Medicines and reported problems in first and last profiles.** (DOCX)

**S3 File. Interview guides.** (DOCX)

## Acknowledgments

The authors thank the participating care home staff, pharmacists and service users for their willingness to engage and contribute to this project. Thanks are also due to Megan Summers, clinical research associate, PRA Health Sciences, Swansea, for data handling, and the project steering group for advice.

## Author Contributions

**Conceptualization:** Sue Jordan.

**Data curation:** Sue Jordan, Hayley Prout, Neil Carter, John Dicomidis, Jamie Hayes.

**Formal analysis:** Sue Jordan, Hayley Prout, Jamie Hayes, Jeffrey Round.

**Funding acquisition:** Sue Jordan, Hayley Prout, Jamie Hayes, Jeffrey Round, Andrew Carson-Stevens.

**Investigation:** Sue Jordan, Hayley Prout, John Dicomidis.

**Methodology:** Sue Jordan, Jeffrey Round.

**Project administration:** Sue Jordan, Hayley Prout, Neil Carter, John Dicomidis, Andrew Carson-Stevens.

**Resources:** Neil Carter, John Dicomidis, Andrew Carson-Stevens.

**Software:** Neil Carter.

**Supervision:** Sue Jordan, Jamie Hayes.

**Validation:** John Dicomidis.

**Visualization:** John Dicomidis, Andrew Carson-Stevens.

**Writing – original draft:** Sue Jordan.

**Writing – review & editing:** Hayley Prout, Neil Carter, John Dicomidis, Jamie Hayes, Jeffrey Round, Andrew Carson-Stevens.

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
