## [Decision Letter · Decision Letter 0]

16 Nov 2020

PONE-D-20-32601

Nobody ever questions – polypharmacy in care homes: a mixed methods evaluation of a multidisciplinary medicines optimisation initiative

PLOS ONE

Dear Dr. Jordan,

Thank you for submitting your manuscript to PLOS ONE. After careful consideration, we feel that it has merit but does not fully meet PLOS ONE’s publication criteria as it currently stands. Therefore, we invite you to submit a revised version of the manuscript that addresses the points raised during the review process.

We look forward to receiving your revised manuscript.

Kind regards,

Prof, Mojtaba Vaismoradi, PhD, MScN, BScN

Academic Editor

PLOS ONE

Journal Requirements:

2.) Please include additional information regarding the interview guide or script used in the study and ensure that you have provided sufficient details that others could replicate the analyses. For instance, if you developed a guide as part of this study and it is not under a copyright more restrictive than CC-BY, please include a copy, in both the original language and English, as Supporting Information.

3.)We note that you have indicated that data from this study are available upon request. PLOS only allows data to be available upon request if there are legal or ethical restrictions on sharing data publicly. For information on unacceptable data access restrictions, please see http://journals.plos.org/plosone/s/data-availability#loc-unacceptable-data-access-restrictions.

Reviewers' comments:

Reviewer's Responses to Questions

**Comments to the Author**

Reviewer #1: Dear authors

I would like to thank you for giving me the opportunity to review this high-quality manuscript. This work is well designed and well written manuscript that improve patient’s safety in home care using the involve of nurses in the Adverse Drug Reaction Profile for polypharmacy. I have some few comments:

In introduction- can you give more literature about using of ADRe-p in health care professions particularly in nurses.

In methods- line 125: please enter the full name of the abbreviation.

Line 142: Is there range for polypharmacy?? In the introduction you report a statistic about polypharmacy in which > 4 was used for polypharmacy. Why you used >3 for polypharmacy? please provide more clarification.

What is difference between Line 142 as a inclusion criteria and line 147 as a exclusion criteria?? In my opinion both of them are same. Please use only a definite inclusion criterion for this item.

In results- Figure 2a and 2b: arrows are crooked.

Reviewer #2: Thank you for the opportunity to review this study. The structure of the manuscript is well put together and it is written up clearly. The only challenge to the manuscript is that it needs a bit of revising for punctuation, inclusion criteria, and discussion section.

- Please, pay attention to punctuation. For instance in lines 268,284 and so on.

- Inclusion criteria: state rationale for why you chose >3 prescribed medicines daily.

- Please, at the first paragraph, describe the aim of the study. Then, discuss your results.

---

## [Author Response · Author response to Decision Letter 0]

3 Dec 2020

Response to reviewers

Nobody ever questions – polypharmacy in care homes: a mixed methods evaluation of a multidisciplinary medicines optimisation initiative

Short title: The adverse drug reaction profile for polypharmacy

Rebuttal letter / response to reviewers [PONE-D-20-32601] - [EMID:148862874e6b707a]

24.11.20 

Dear Professor Vaismoradi,

Thank you for the return of our manuscript. We are very grateful for the reviewers’ support and suggestions. We have done our best to comply, as tabulated below. 

We feel that clinicians need to be aware of the importance of monitoring patients for any adverse effects of prescribed medicines. Publication in PlosOne would stimulate prescribers to take a proactive approach to medication review for people in long-term care. 

We hope that readers will concur with the reviewers: 

The work has not been submitted elsewhere. 

Thank you for your time spent on this paper. 

Yours,

Sue Jordan, on behalf of all authors

Changes Made

We have tracked in the text where we have made the suggested changes. The paper has been submitted with and without tracks. 

Journal Requirements:

 We have done our best to comply. We are unaware of any outstanding issues, but are happy to modify if required. 

2.) Please include additional information regarding the interview guide or script used in the study and ensure that you have provided sufficient details that others could replicate the analyses. For instance, if you developed a guide as part of this study and it is not under a copyright more restrictive than CC-BY, please include a copy, in both the original language and English, as Supporting Information.

This has been appended as supplementary material, S3 file. 

3.)We note that you have indicated that data from this study are available upon request. PLOS only allows data to be available upon request if there are legal or ethical restrictions on sharing data publicly. For information on unacceptable data access restrictions, please see http://journals.plos.org/plosone/s/data-availability#loc-unacceptable-data-access-restrictions.

The Data Availability Statement now reads: 

Data are provided within the supplementary tables. Ethical restrictions have been imposed on data sharing by the NHS Research Ethics Committee that approved this study. The data contain potentially identifying and sensitive information. The data used in this study are available to the research data community at https://zenodo.org/record/4090384#.X4hKptZFzLZ Swansea University, Swansea, UK. 

All proposals to view the data are subject to review by Swansea University’s Research Governance department and the PI. Before any data can be accessed, approval must be given. 

The application process is via the Academic Lead for Research Integrity Research Engagement & Innovation Services, Swansea University and the PI or Neil Carter.

Contacts: Swansea University, Swansea SA2 8PP

• Tel: +44 /0 1792 606060 and 518541 or 295610

• Email: researchgovernance@swansea.ac.uk, s.e.jordan@swansea.ac.uk or n.carter@swansea.ac.uk

The research instrument used in the study is available for clinical use without charge via the project website: http://www.swansea.ac.uk/adre/

 Our data are restricted, as above.

 Response to reviewers

We are very grateful for the detailed consideration given to our paper, and have responded to the points raised below.

Reviewer #1: 

Dear authors, 

I would like to thank you for giving me the opportunity to review this high-quality manuscript. This work is well designed and well written manuscript that improve patient’s safety in home care using the involve of nurses in the Adverse Drug Reaction Profile for polypharmacy. I have some few comments: Thank you for your support.

In introduction- can you give more literature about using of ADRe-p in health care professions particularly in nurses. We have moved and added some text on this to the introduction, last sentence:

Our intervention, the ADRe Profile [2-4], achieves this by uniting the tacit, experiential knowledge of nurses and carers with records of the patients’ clinical problems and possible causes, in a form that can be shared within the multidisciplinary team [2-7] (Fig 1). 

This entails moving Fig 1. 

In methods- line 125: please enter the full name of the abbreviation. Standards for Quality Improvement Reporting Excellence – done

Line 142: Is there range for polypharmacy?? In the introduction you report a statistic about polypharmacy in which > 4 was used for polypharmacy. Why you used >3 for polypharmacy? please provide more clarification. Definitions differ. We explain in the inclusion criteria: 

This definition of polypharmacy is used by Cochrane reviewers [27]. We were unable to adopt definitions based on appropriateness or duration of prescriptions before gaining consent to view records.

What is difference between Line 142 as a inclusion criteria and line 147 as a exclusion criteria?? In my opinion both of them are same. Please use only a definite inclusion criterion for this item. The line is deleted – thank you.

In results- Figure 2a and 2b: arrows are crooked. The figures have been redrawn and new tif files uploaded.

Reviewer #2: Thank you for the opportunity to review this study. The structure of the manuscript is well put together and it is written up clearly. The only challenge to the manuscript is that it needs a bit of revising for punctuation, inclusion criteria, and discussion section. 

Thank you

- Please, pay attention to punctuation. For instance in lines 268,284 and so on. Line 268, now 269, ‘and’ has been substituted for ‘or’.

Line 284 – no errors identified.

Line 455, now 468 – full stop substituted for semicolon

Line 458, now 459 – definite article removed

Line 460, now 464 – ‘however’ substituted for ‘but’

Arabic numerals replaced with text throughout. 

- Inclusion criteria: state rationale for why you chose >3 prescribed medicines daily. We added this statement, as above:

This definition of polypharmacy is used by Cochrane reviewers [27]. We were unable to adopt definitions based on appropriateness or duration of prescriptions before gaining consent to view records.

- Please, at the first paragraph, describe the aim of the study. We have added to start of discussion:

We aimed to identify changes following nurses’, carers’ and pharmacists’ use of ADRe-p.

---

## [Editor Report · Decision Letter 1]

11 Dec 2020

Nobody ever questions – polypharmacy in care homes: a mixed methods evaluation of a multidisciplinary medicines optimisation initiative

PONE-D-20-32601R1

Dear Dr. Jordan,

We’re pleased to inform you that your manuscript has been judged scientifically suitable for publication and will be formally accepted for publication once it meets all outstanding technical requirements.

Kind regards,

Prof, Mojtaba Vaismoradi, PhD, MScN, BScN

Academic Editor

PLOS ONE

---

## [Editor Report · Acceptance letter]

21 Dec 2020

PONE-D-20-32601R1 

Nobody ever questions – polypharmacy in care homes: a mixed methods evaluation of a multidisciplinary medicines optimisation initiative 

Dear Dr. Jordan:

I'm pleased to inform you that your manuscript has been deemed suitable for publication in PLOS ONE. Congratulations! Your manuscript is now with our production department. 

Kind regards, 

on behalf of

Professor Mojtaba Vaismoradi 

Academic Editor

PLOS ONE